# Distribution Discrepancy and Feature Heterogeneity for Active 3D Object Detection

**Huang-Yu Chen**[1]  **Jia-Fong Yeh**[1]  **Jia-Wei Liao**[1]  **Pin-Hsuan Peng**[1]  **Winston H. Hsu**[1,2]

[1]National Taiwan University    [2]MobileDrive

**Abstract:** LiDAR-based 3D object detection is a critical technology for the development of autonomous driving and robotics. However, the high cost of data annotation limits its advancement. We propose a novel and effective active learning (AL) method called Distribution Discrepancy and Feature Heterogeneity (DDFH), which simultaneously considers geometric features and model embeddings, assessing information from both the instance-level and frame-level perspectives. Distribution Discrepancy evaluates the difference and novelty of instances within the unlabeled and labeled distributions, enabling the model to learn efficiently with limited data. Feature Heterogeneity ensures the heterogeneity of intra-frame instance features, maintaining feature diversity while avoiding redundant or similar instances, thus minimizing annotation costs. Finally, multiple indicators are efficiently aggregated using Quantile Transform, providing a unified measure of informativeness. Extensive experiments demonstrate that DDFH outperforms the current state-of-the-art (SOTA) methods on the KITTI and Waymo datasets, effectively reducing the bounding box annotation cost by 56.3% and showing robustness when working with both one-stage and two-stage models. Source code: *https://github.com/Coolshanlan/DDFH-active-3Ddet*

**Keywords:** Active Learning, LiDAR 3D Object Detection, Autonomous Driving

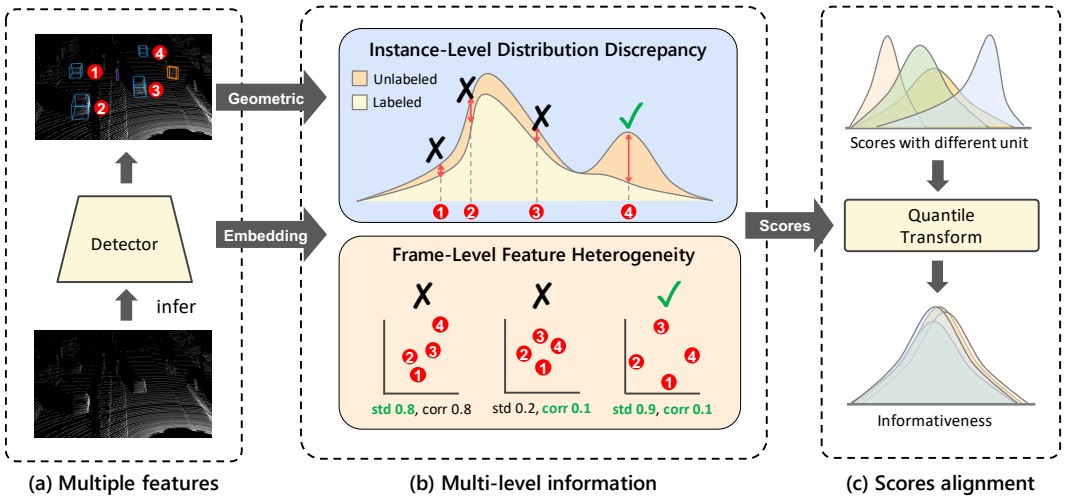

Figure 1: The three core concepts of DDFH. (a) Embedding and geometric features are used as DDFH inputs. (b) Considering instance-level distribution discrepancy and frame-level feature heterogeneity ensures that instances remain highly informative across all levels. (c) After transforming various indicators using the Quantile Transform, it effectively aggregates to estimate the final informativeness.

8th Conference on Robot Learning (CoRL 2024), Munich, Germany.

# 1   Introduction

Research on LiDAR-based 3D object detection [1, 2] has emerged due to its significant potential and applications. However, annotating LiDAR data requires locating multiple objects in three-dimensional space, which is expensive and time-consuming. Therefore, obtaining annotated data more efficiently within limited time and resources has become a crucial and unavoidable issue. Several studies have attempted to reduce annotation costs through Auto Labeling [3, 4] or Domain Adaptation [5, 6], which rely on a small number of annotated samples. However, unstable annotation accuracy and restricted domain shifts may limit their usage in various applications.

Active Learning (AL), aiming to select the most informative samples from a large pool of unlabeled data for human annotation, is a significant solution to reduce the model's dependence on the amount of data. Although AL has been proven effective in mitigating annotation costs in various research domains [7, 8], its application to LiDAR-based 3D Object Detection is underexplored, with three main challenges remaining unresolved: (1) Compared to 2D object detection, LiDAR-based object detection has additional geometric features (such as rotation and point density) that need to be considered. (2) While the detector's predictions are instance-level, the selection in AL is frame-level, making it challenging to propagate from instances to frames and estimate informativeness. (3) Aggregating multiple indicators under different units and scales is also challenging.

Previous work has used general AL strategies such as entropy and ensemble methods to estimate uncertainty. However, they neglected the unique geometric features in LiDAR-based object detection (first challenge), and solely assessing the instance-level informativeness is insufficient to address the second challenge. A recent work, CRB [9], proposed three heuristic methods to estimate label balance, representativeness, and point density balance through a staged filtering approach. Unfortunately, the filtering order significantly affects sampling results, impacting the fairness among indicators and failing to solve the third challenge. Another work, KECOR [10], proposed a kernel coding rate maximization strategy. However, it also did not consider geometric features and used different weighted settings to aggregate multiple indicators in different datasets, affecting generalization.

To address the above challenges, we propose the Distribution Discrepancy and Feature Heterogeneity (DDFH) method, as illustrated in Fig.1, where components (a), (b), and (c) are designed for the first, second, and third challenges, respectively. DDFH leverages model embedding and geometric information as features for informativeness estimation to address the first challenge. Moreover, we explores informativeness from instance-level and frame-level perspectives by considering intra-class **Distribution Discrepancies** (DD) and intra-frame **Feature Heterogeneity** (FH) to tackle the second challenge. Then, DDFH employs a **Quantile Transform** (QT) to normalize each indicator to the same scale, effectively aggregating the indicators to solve the third challenge. Finally, we propose **Confidence Balance** (CB) to evaluate the allocation of annotation resources. Unlike previous methods that solely count selected instances for each category, CB considers the summation of confidence levels for each instance within the same category.

We verify the effectiveness of DDFH through experiments on real-world datasets, KITTI and Waymo Open Dataset. The results indicate that our DDFH method outperforms the existing SOTA, effectively reducing the data annotation cost by 56.3% and achieving an average improvement of 1.8% in 3D mAP with the same amount of data. From our extensive ablation studies, DDFH also demonstrates its generalization when used with both one-stage and two-stage detection models.

# 2   Related Work

**LiDAR-based 3D Object Detection.** LiDAR-based object detection techniques are primarily divided into two categories: point cloud direct processing and voxelization. Methods such as the PointNet series [11, 12] operate directly on point clouds, preserving the original spatial accuracy of the data but are less efficient in handling large-scale data. Recent research, like PointAugmenting [13], introduces cross-modal augmentation, enhancing LiDAR point clouds with deep features extracted from pre-trained 2D object detection models, thereby improving 3D object detection per-

formance. Voxelization methods, such as VoxNet [14] and SECOND [15], convert point clouds into voxel grids to enable efficient 3D convolution, significantly increasing computational speed. The Voxel Transformer (VoTr) [16] architecture effectively expands the model's receptive field, enhancing the ability to capture large-scale environmental information. The PV-RCNN series [17, 18] improves detection accuracy and processing efficiency by fusing point cloud and voxel features. These detectors rely heavily on large volumes of high-quality training data; however, the labeling cost for LiDAR data is quite expensive.

**Active learning for object detection.** Active learning selects the most informative samples for annotation, thus mitigating the model's dependence on the volume of data. Numerous general active learning strategies currently exist, such as those based on model uncertainty [4, 19, 20, 21, 22], diversity [23, 24, 25], or hybrid methods that combine both approaches [26]. Estimating in the gradient space is also a common approach (e.g., BADGE [27], BAIT [28]). Research applying these methods to object detection [29, 30, 31, 32] remains limited. Many studies directly use general strategies like maximum entropy [33], bayesian inference [34] to estimate uncertainty in both bounding box and category. LT/C [35] introduces noise-perturbed samples and assesses tightness and stability based on the model's output. The estimation of information quantity through the output probability distribution is also a common approach [36, 37]. Research on LiDAR-based object detection is even scarcer, mainly due to the high computational cost of point cloud processing and the higher dimensionality of regression information. General AL methods (Shannon Entropy [38], ensemble [29]) do not consider geometric features and therefore are not well-suited for LiDAR. Recent work, such as CRB [9], proposes three heuristic methods to incrementally filter samples. KECOR [10] identifies the most informative samples through the lens of information theory. However, these studies fail to effectively integrate multi-level information and do not consider the distribution of selected samples, leading to redundant annotation costs. Therefore, we propose DDFH, which estimates the informativeness based on the distributional differences of instances and intra-frame feature heterogeneity, and effectively aggregates multiple indicators using the quantile transform to estimate multi-level informativeness.

## 3 Methodology

### 3.1 Active Learning Setup

In the active object detection setup, the labeled set $D^L = \{(P^L, Y^L)\}$ contains a small amount of point clouds $P^L$ with annotations $Y^L$, and $D_U = \{P_U\}$ represents a large unlabeled set of raw point clouds $P_U$. Initially, samples are randomly selected to form $D_L$, and the detection model learns over multiple rounds $r \in \{1, ..., R\}$. The objective of active learning is to evaluate the informativeness of $P_U$ in each round and select the most informative samples to form a new subset $D_s^*$ for human annotation. Then, the $D_s^*$ is merged into $D_L$ to start a new round to retrain the model. This process repeats until the size of the labeled set reaches the annotation budget.

### 3.2 Framework Overview

We propose a novel active learning framework, Distribution Discrepancy and Feature Heterogeneity (DDFH) for LiDAR-based 3D object detection. As illustrated in Fig. 2, we infer point clouds $P \in \{D_U, D_L\}$ into the model and get model output containing embeddings and bounding boxes. However, estimating distributions in high-dimensional space is challenging, so we use t-SNE[40] to project embeddings into lower dimensions while retaining important information, denoted as $\mathbf{f}^{e*} \in \mathbb{R}^2$. The geometric features of LiDAR-based object detection (length, width, height, volume, rotation, and point cloud density) $\mathbf{f}^g \in \mathbb{R}^6$ are also significant, as they convey direct information about objects such as occlusion, behavior, and morphology. So we use $\mathbf{f} = [\mathbf{f}^{e*\top} \ \mathbf{f}^{g\top}]^\top \in \mathbb{R}^8$ as the input feature for DDFH, calculating multiple indicators to estimate informativeness. However, since the units of the indicators differ, normalization is required before aggregation.

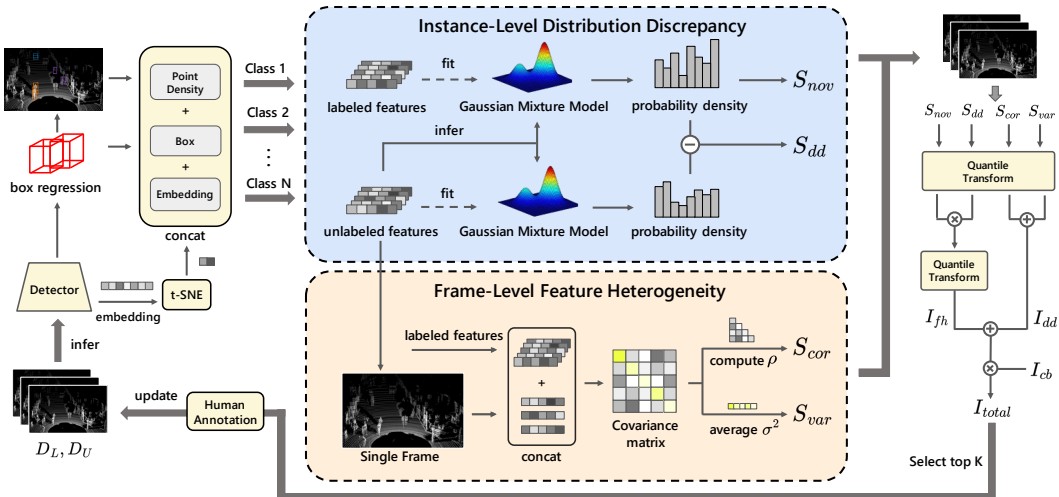

Figure 2: DDFH framework for LiDAR-based 3D active object detection. According to the batch active learning[39] setup, one cycle represents a single sampling. DDFH utilizes a Quantile Transform to normalize all metrics before aggregation to estimate informativenes, and then updates the dataset before starting a new round.

**Score Normalization.** DDFH evaluates informativeness based on multiple indicators, but their scales differ, making direct aggregation impossible. Thus, we use Quantile Transform $\psi$ to normalize. $\psi(.,.)$ is a non-linear transform that converts the first input to follow a normal distribution and returns the transformed result of the second input. $\psi$ spreads out the most frequent values and reduces the impact of outliers. Since the goal of active learning is to select the top k samples, the relative distance of scores is not particularly important. Instead, maintaining the ranks of various indicators and reducing outliers aids in aggregating the indicators, making $\psi$ a crucial bridge in computing $I_{total}$. Next, we will introduce the operating principles of DDFH sequentially.

### 3.3 Instance-Level Distribution Discrepancy

Since labeled set is much smaller than the unlabeled set, estimating the distribution is particularly important. Reducing the distribution gap between the labeled set and the unlabeled set will assist the model in inference. Inspired by [37], we use a Gaussian Mixture Model(GMM) to estimate probability density. Unlike previous works, we consider both geometric features and embeddings, using dimensionality reduction to avoid overly sparse space. Intuitively, if an instance appears frequently in the unlabeled set but is rare in the labeled set, such an instance can help the model efficiently understand unlabeled samples. Therefore, for each class $c \in C$, we establish $G_c^L$ fit on $\{\mathbf{f_{c,1}}...\mathbf{f_{c,N_c^L}}\}$ and $G_c^U$ fit on $\{\mathbf{f_{c,1}}...\mathbf{f_{c,N_c^U}}\}$, where $G_c^U$ is a GMM fit on unlabeled features, and $N_c^U$ is the number of instances with class c in the unlabeled set. We estimate the probability density function of each instance in the unlabeled and labeled sets, and calculate the discrepancy score $s_i^{dd}$:

$$s_{c,i}^{dd} = \mathbb{P}_{G_c^U}(\mathbf{f}_{c,i}) - \mathbb{P}_{G_c^L}(\mathbf{f}_{c,i}), \quad i = 1, 2, ..., N_c^U, \tag{1}$$

where $\mathbb{P}_{G_c^U}(\mathbf{f}_{i,c})$ represents the probability density of $\mathbf{f}_{i,c}$ in the unlabeled set. However, considering $S_{dd}$ alone is insufficient, as very dense instances might overly influence the indicator, leading to frequent selection of dense but repetitive instances in the early stages. Hence, unlike previous works [7], we also extract $\mathbb{P}_{G_c^L}(\mathbf{f}_{i,c})$ to calculate the novelty score $s^{nov}$:

$$s_{c,i}^{nov} = -\mathbb{P}_{G_c^L}(\mathbf{f}_{c,i}), \quad i = 1, 2, ..., N_c^U. \tag{2}$$

$s^{nov}$ ensures the novelty of instances. If an instance has a high probability density in the labeled set, it indicates that the instance has already been selected, thus the $s^{nov}$ score will decrease, effectively reducing redundant annotation costs. Following these two indicators, $I_{dd}(P_j)$ can be calculated as:

$$I_{dd}(P_j) = \frac{1}{N} \sum_{i=1}^{N} [\psi(S^{dd}, s_i^{dd}) + \psi(S^{nov}, s_i^{nov})], \tag{3}$$

where $N$ is the number of instances in $P_j$. $S^{dd}$ is the set of all instances' $s^{dd}$. Ablation studies show that $I_{dd}$ enables the active learning model to learn rapidly with a small amount of data. However, $I_{dd}$ does not consider frame-level information, which may overlook similar instances within the same frame. Therefore, we introduce a second component to address this issue.

### 3.4 Frame-Level Feature Heterogeneity

Object detection is a multiple-instance problem. Considering only instance-level information can lead to redundant annotations. Complex scenes usually contain numerous objects that share the same lighting and environmental factors, making their features highly similar or following linear variations. Such samples are costly to annotate but offer limited assistance to the model. Therefore, we propose frame-level Feature Heterogeneity (FH). As shown in Fig. 1b, we decompose heterogeneity into correlation and variance. Denote $F_{j,c}^U = [\mathbf{f}_{j,c,1}^U ... \mathbf{f}_{j,c,m}^U]$ as the feature vector of all instances with class c in j-frame, where $m$ represents the number of instances. FH ensures $F_{j,c}$ can maximize the feature heterogeneity of the labeled instance vector $F_c^L$. Specifically, we combine the two into $\tilde{F}_{j,c} = [F_{j,c}^U \, F_c^L] \in \mathbb{R}^{8 \times N}$ where $N$ as the number of all sample. We use covariance $cov$ and variance $\sigma^2$ to calculate the Pearson correlation $\rho$, ensuring non-linear variations among features. The correlation $\rho$ is calculated as:

$$\rho(\tilde{F}_{j,c}) = \frac{2}{N_f(N_f - 1)} \sum_{k<\ell}^{N_f} \frac{cov(\hat{F}_{j,c}^k, \tilde{F}_{j,c}^\ell)}{\sigma(\tilde{F}_{j,c}^k) \cdot \sigma(\tilde{F}_{j,c}^\ell)}, \qquad (4)$$

where $\tilde{F}_{j,c}^k$ and $\tilde{F}_{j,c}^\ell$ represent the $k$-th and $\ell$-th feature dimension of matrix $\tilde{F}_{j,c}$, and $\overline{\tilde{F}_{j,c}^k}$ is the mean of $\tilde{F}_{j,c}^k$. $\sigma^2(\tilde{F}_{j,c}^k) = \frac{1}{N}\|\tilde{F}_{j,c}^k - \overline{\tilde{F}_{j,c}^k}\|_2^2$, $N_f$ is the number of feature dimensions of matrix $\tilde{F}_{j,c}$. The smaller the value of $\rho$, the less linear the correlation, enabling the model to learn more feature combinations. However, as shown in Fig. 1, considering only correlation overlooks the information about the distance between features. Thus, we also consider $\sigma^2$ to ensure sufficient variation among features. Based on these two indicators, we calculate $s^{cor}$ and $s^{var}$:

$$s_{j,c}^{var} = \sum_{k=1}^{N_f} \sigma^2(\tilde{F}_{j,c}^k), \quad s_{j,c}^{cor} = 1 - |\rho(\tilde{F}_{j,c})|. \qquad (5)$$

The closer $\rho$ is to 0, the less linear the correlation between features. Conversely, values far from 0 indicate positive or negative correlations. Therefore, we take the absolute value of correlation and subtract it from 1, making the $s_{j,c}^{cor}$ indicator larger. Based on these two scores, we calculate the feature heterogeneity informativeness $I_{fh}(P_j)$ for $P_j$:

$$I_{fh}(P_j) = \frac{1}{C} \sum_{c=1}^{C} \psi(S_{var}, s_{j,c}^{var}) \cdot \psi(S_{cor}, s_{j,c}^{cor}) \qquad (6)$$

where $S^{var}$ is the set of all $s^{dd}$. $I_{fh}$ ensures instances maintain low correlation and high variance, calculated through multiplication. This introduces the core components of the DDFH approach, exploring informativeness from both instance-level and frame-level perspectives.

### 3.5 Confidence Balance for Imbalanced Data

Balancing annotation costs across classes has always been crucial in active learning. Previous works [10, 9] calculate entropy by the number of all categories and the classification logit from the classifier, termed Label Balance (LB). However, these method is limited in imbalanced datasets, as imbalanced classes usually have lower confidence, resulting in more false-positive instances. The actual quantity is often less than expected. Therefore, we propose Confidence Balance (CB) $I_{cb}$, replacing instance quantities with the sum of confidences in each category to better reflect the true class distribution in the frame, enhancing the number of minority classes. $I_{cb}$ can be calculated as follows:

$$I_{cb}(P_j) = -\sum_{c=1}^{C} \phi(p_{j,c}) \log \phi(p_{j,c}), \quad \phi(p_{j,c}) = \frac{e^{p_{j,c}}}{\sum_{c=1}^{C} e^{p_{j,c}}}, \qquad (7)$$

Table 1: Compare 3D mAP(%) scores for general AL and AL for detection in KITTI Dataset with two-stage 3D detector PV-RCNN

| | Method | AVERAGE | | | CAR | | | PEDESTRIAN | | | CYCLIST | | |
|---|---|---|---|---|---|---|---|---|---|---|---|---|---|
| | | Easy | Mod. | Hard | Easy | Mod. | Hard | Easy | Mod. | Hard | Easy | Mod. | Hard |
| Generic | CORESET [24] | 72.26 | 59.81 | 55.59 | 87.77 | 77.73 | 72.95 | 47.27 | 41.97 | 38.19 | 81.73 | 59.72 | 55.64 |
| | BADGE [41] | 75.34 | 61.44 | 56.55 | 89.96 | 75.78 | 70.54 | 51.94 | 40.98 | 45.97 | 84.11 | 62.29 | 58.12 |
| | LLAL [42] | 73.94 | 62.95 | 58.88 | 89.95 | 78.65 | 75.32 | 46.94 | 45.97 | 45.97 | 75.55 | 60.35 | 55.36 |
| AL Detection | LT/c [35] | 75.88 | 63.23 | 58.89 | 88.73 | 78.12 | 74.87 | 55.17 | 48.37 | 43.63 | 83.72 | 63.21 | 59.16 |
| | Mc-REG [9] | 66.21 | 54.41 | 51.70 | 88.85 | 76.21 | 73.87 | 35.82 | 31.81 | 29.79 | 73.98 | 55.23 | 51.85 |
| | Mc-MI [38] | 71.19 | 57.77 | 53.81 | 86.28 | 75.58 | 71.56 | 41.05 | 37.50 | 33.83 | 86.26 | 60.22 | 56.04 |
| | CONSENSUS [29] | 75.01 | 61.09 | 57.60 | 90.14 | 78.01 | 74.28 | 56.43 | 49.50 | 44.80 | 78.46 | 55.77 | 53.73 |
| | CRB [9] | 79.06 | 66.49 | 61.76 | 90.81 | 79.06 | 74.73 | 62.09 | 54.56 | 48.89 | 84.28 | 65.85 | 61.66 |
| | CRB(offi.) | 80.70 | 67.81 | 62.81 | 90.98 | 79.02 | 74.04 | 64.17 | 50.82 | 50.82 | 86.96 | 67.45 | 63.56 |
| | KECOR [10] | 79.81 | 67.83 | 62.52 | 91.43 | 79.63 | 74.41 | 63.49 | 56.31 | 50.20 | 84.51 | 67.54 | 62.96 |
| | KECOR(offi.) | 81.63 | 68.67 | 63.42 | 91.71 | 79.56 | 74.05 | 65.37 | 57.33 | 51.56 | 87.80 | 69.13 | 64.65 |
| | DDFH(Ours) | **82.27** | **69.84** | **64.76** | **91.76** | **80.65** | **76.46** | **66.37** | **59.40** | **52.97** | **88.68** | **69.47** | **64.85** |

Figure 3: 3D mAP(%) of DDFH and AL baselines on the KITTI val split with PV-RCNN.

where $p_{j,c}$ represents the sum of confidences of all instances of class c in the j-th frame. We compare the effectiveness of LB and CB sampling in Fig. 4(c-d), demonstrating the importance of $I_{cb}$ for imbalanced data.

## 3.6 Acquisition Function

As described in 3.2, DDFH combines multiple indicators to explore informativeness comprehensively. Through QT, all indicators are normalized to compute a unified informativeness indicator $I_{total}$, ensuring that the top-k frames are efficient, novel, heterogeneous, and balanced, identifying the optimal selected sets $D_s^*$, formulated as:

$$D_s^* = \underset{D_s \subset D_U}{\arg\max} \quad I_{total}(P_U), \ I_{total}(P_j) = (I_{dd}(P_j) + I_{fh}(P_j)) \cdot I_{cb}(P_j). \tag{8}$$

$I_{dd}$ and $I_{fh}$ evaluate informativeness, and multiplying by $I_{cb}$ ensures balanced annotation costs across classes while considering informativeness.

## 4 Experiments

### 4.1 Experimental Settings

**3D Point Cloud Datasets.** We tested our method on two real-world datasets: KITTI [43] and Waymo Open Dataset [44]. KITTI contains approximately 7,481 training point clouds (3712 for training, 3769 for validation) with annotations. Each point cloud is annotated with 3D bounding boxes for cars, pedestrians, and cyclists, totaling 80,256 objects. The Waymo Open Dataset provides a large-scale collection of data, it contains 158,361 training point clouds and 40,077 testing point clouds. The sampling intervals for KITTI and Waymo are set to 1 and 10, respectively.

**Baselines.** We comprehensively evaluated 6 general active learning (AL) methods and 6 AL methods for object detection. RAND selects samples randomly. ENTROPY [45] and LLAL [42] are uncertainty-based methods. CORESET [24] is a diversity-based method. BAIT [28] and BADGE [41] are hybrid methods. MC-MI [38] and MC-REG [9] utilize Bayesian inference. CONSENSUS [29] employs an ensemble to calculate the consensus score. LT/C evaluates instability and local-

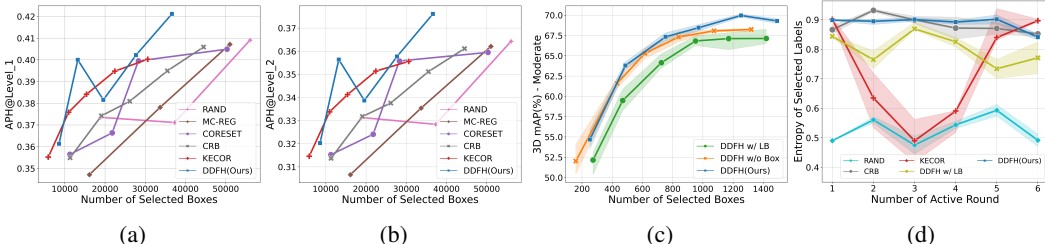

Figure 4: (a-b) report 3D APH of various AL methods on different difficulties in the Waymo dataset. (c-d) are experiments on the KITTI dataset. (c) demonstrates the impact on performance when DDFH omits geometric features and replaces confidence balance with label balance. (d) calculates the entropy of the number of samples selected for each class to compare the effectiveness of different AL methods in balancing annotation costs.

ization tightness. CRB [9] progressively filters based on three heuristic methods. KECOR [10] identifies the most informative sample through the lens of information theory.

**Evaluation Metrics.** We follow the work of KECOR [10]: In the KITTI dataset, we utilize Average Precision (AP) to evaluate object location in Bird Eye View and 3D, calculated with 40 recall positions. The task difficulty is categorized as EASY, MODERATE (MOD.), and HARD based on the visibility, size, and occlusion/ truncation of the objects. In Waymo, we use Average Precision with Heading (APH), which incorporates both bounding box overlap and orientation accuracy. It categorizes objects into Level 1 (at least five LiDAR points, easier to detect) and Level 2 (all objects, including more challenging scenarios).

**Implementation Details.** We strive to avoid excessive parameter tuning by using a unified set of hyper-parameters across all experiments. We set the perplexity for t-SNE to 100. For Gaussian Mixture Model, set number of components to 10, reg_covar to 1e-2 to increase generalization and initialize parameters by k-means++.

## 4.2 Main Results

Since the most of baseline method do not provide initial selection samples, to present the results more fairly, we reproduce most of the baseline methods using the same initial settings and provided the official reported performance of two recent methods (CRB and KECOR) in our experiments.

**DDFH with Two-Stage Detection Model.** For the KITTI dataset, We ran each method three times. As show in Fig. 3, our method outperforms all baseline methods. Compared with the CRB and KECOR, the annotation cost is reduced by 51.7% and 33.2%, respectively, especially when the number of annotations is small, the growth rate is particularly fast. In Table 1, we follow the KECOR setting, showing the performance with 800 (1%) bounding box annotations. Under the same initialization conditions, our average performance surpasses the SOAT by 2.23%, especially improving by 4.17% in the imbalanced class (Cyclist). For the Waymo Dataset, in Fig. 4(a-b), the Level-1 and Level-2 performance are reported, respectively. Compared to KECOR and CRB, DDFH improves APH by 1.8% and 3.8%, respectively, and reduces the annotation cost by 56.3% and 66.4%, respectively, demonstrating the effectiveness of DDFH in more diverse and complex scenarios.

**DDFH with One-Stage Detection Model.** Table 2 reports the 3D and BEV mAP scores with 1% annotation bounding boxes. Compared to the SOTA, it improves by approximately 2.8% in 3D mAP and 2.28% in BEV mAP. In Fig. 5, we further report the performance growth trend for each category in different levels of difficulty in MOD. DDFH, especially in the car category, which often leads to excessive annotations, can quickly improve performance with the most streamlined annotation cost. For the pedestrian category, the uncertainty-based method performs exceptionally well with a small number of annotations. However, the lack of consideration for diversity limits the performance.

## 4.3 Ablation Study

Table 2: Compare 3D mAP and BEV scores for general AL and AL for detection in KITTI Dataset with one-stage 3D detector SECOND

| Method | Venue | 3D Detection mAP | | | BEV Detection mAP | | |
|---|---|---|---|---|---|---|---|
| | | EASY | MOD. | HARD | EASY | MOD. | HARD |
| RAND | | 66.67 | 53.15 | 49.12 | 72.65 | 60.94 | 57.12 |
| ENTROPY [45] | IJCNN'14 | 69.29 | 56.58 | 51.59 | 74.52 | 63.38 | 58.65 |
| BALD [21] | ICML'17 | 68.78 | 55.49 | 50.30 | 74.21 | 62.51 | 57.70 |
| CORESET [24] | ICLR'18 | 65.96 | 51.79 | 47.65 | 73.85 | 60.22 | 56.06 |
| LLAL [42] | CVPR'19 | 68.51 | 56.05 | 50.97 | 74.57 | 63.64 | 58.92 |
| BADGE [41] | ICLR'20 | 69.09 | 55.20 | 50.72 | 75.12 | 63.05 | 58.81 |
| BAIT [28] | NeurIPS'21 | 69.45 | 55.61 | 51.25 | 76.04 | 63.49 | 53.40 |
| CRB [9] | ICLR'23 | 71.69 | 57.16 | 52.35 | 78.01 | 64.71 | 60.04 |
| KECOR [10] | ICCV'23 | 71.85 | 57.75 | 52.56 | 78.30 | 65.41 | 60.15 |
| DDFH(Ours) | | **74.13** | **60.61** | **55.48** | **79.65** | **67.95** | **63.10** |

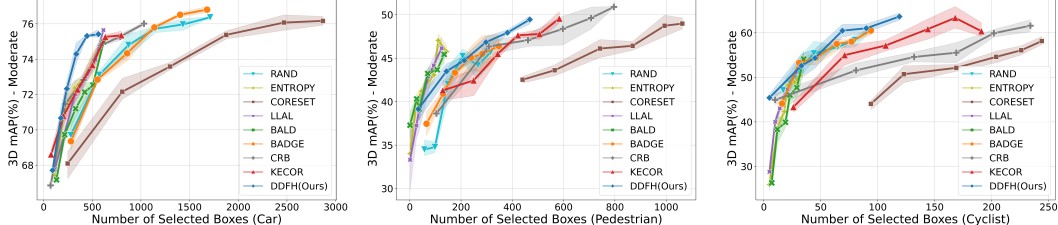

Figure 5: 3D mAP(%) of DDFH and the AL Baseline across various categories on the KITTI dataset at the moderate difficulty with SECOND.

**Efficacy of Confidence Balance.** As shown in Table 3, CB effectively improves the 3D mAP of the imbalanced class (Cyclist) by 4.3% compared to LB. From Fig. 4c, it is evident that the performance using LB decreases by an average of 2.8% 3D mAP in each round. In Fig. 4d, we further demonstrate the label entropy of the selected samples. DDFH can effectively allocate annotation resources in each round, while KECOR sacrifices balance while considering informativeness.

Table 3: Compare the impact of each component on 3D mAP scores of KITTI dataset at the moderate difficulty

| DD | FH | CB | LB | Average | Car | Pedes. | Cyclist |
|---|---|---|---|---|---|---|---|
| - | - | - | - | 62.97 | 79.44 | 48.93 | 60.52 |
| - | - | - | ✓ | 66.87 | 78.89 | 56.36 | 65.38 |
| - | - | ✓ | - | 67.69 | 78.41 | 54.96 | **69.71** |
| - | ✓ | ✓ | - | 64.73 | 80.15 | 51.09 | 62.94 |
| ✓ | - | ✓ | - | 68.94 | 79.94 | 58.22 | 68.66 |
| ✓ | ✓ | ✓ | - | **69.84** | **80.65** | **59.40** | 69.47 |

**Efficacy of Distribution Discrepancy and Feature Heterogeneity.** The results in Table 3 show that DD can significantly enhance overall performance, especially in categories with fewer instances, such as pedestrians and cyclists, by focusing more on the selection of these categories due to their larger distribution differences. Using FH alone to estimate informativeness without considering the labeled distribution would be too limiting. The experiments demonstrate that DDFH effectively combines the advantages of both components, resulting in improvements across all categories.

**Efficacy of Geometric Features.** In Fig. 4c, the results show that after considering geometric features, DDFH improves the average performance by 1.6% in 3D mAP, confirming that diverse geometric features help the model capture a wider variety of objects.

## 5 Conclusion

We propose a novel active learning framework DDFH for LiDAR-based 3D object detection that integrates model features with geometric characteristics. By exploring point cloud data through instance-level distribution discrepancy and frame-level feature heterogeneity, and introducing confidence balance, we enhance annotations for imbalanced classes. Our extensive experiments show that compared to SOTA, DDHF reduces annotation costs by 56%, improves performance by 1.8%, and efficiently extracts richer information, demonstrating its effectiveness over current methods.

**Acknowledgments**

This work was supported in part by National Science and Technology Council, Taiwan, under Grant NSTC 112-2634-F-002-006. We are grateful to MobileDrive and the National Center for High-performance Computing.

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

# Supplementary Material of Distribution Discrepancy and Feature Heterogeneity for Active 3D Object Detection

Table 1: Compare 3D mAP(%) scores for different SOTA apporch in KITTI Dataset when acquiring approximately 1% queried bounding boxes. [†] indicates the reported performance of the backbone trained with the 100% labeled set.

| Method | AVERAGE | | | CAR | | | PEDESTRIAN | | | CYCLIST | | |
|---|---|---|---|---|---|---|---|---|---|---|---|---|
| | Easy | Mod. | Hard | Easy | Mod. | Hard | Easy | Mod. | Hard | Easy | Mod. | Hard |
| CRB [1] | 79.06 | 66.49 | 61.76 | 90.81 | 79.06 | 74.73 | 62.09 | 54.56 | 48.89 | 84.28 | 65.85 | 61.66 |
| CRB(offi.) | 80.70 | 67.81 | 62.81 | 90.98 | 79.02 | 74.04 | 64.17 | 50.82 | 50.82 | 86.96 | 67.45 | 63.56 |
| KECOR [2] | 79.81 | 67.83 | 62.52 | 91.43 | 79.63 | 74.41 | 63.49 | 56.31 | 50.20 | 84.51 | 67.54 | 62.96 |
| KECOR(offi.) | 81.63 | 68.67 | 63.42 | 91.71 | 79.56 | 74.05 | 65.37 | 57.33 | 51.56 | 87.80 | 69.13 | 64.65 |
| DDFH(Ours) | **82.27** | **69.84** | **64.76** | **91.76** | **80.65** | **76.46** | **66.37** | **59.40** | **52.97** | **88.68** | **69.47** | **64.85** |
| PV-RCNN[†] | 81.75 | 70.99 | 67.06 | 92.56 | 84.36 | 82.48 | 64.26 | 56.67 | 51.91 | 88.88 | 71.95 | 66.78 |

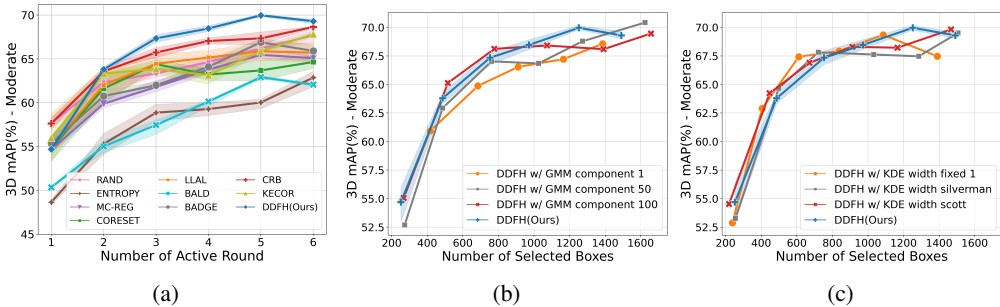

Figure 6: (a) report 3D mAP of various AL methods on KITTI in each active round. (b-c) represents the impact of different density estimation methods and varying parameter settings on the performance of DDFH.

## A  More Implementation Details

To ensure the fairness and reproducibility of our experiments, we implemented DDFH and reproduced most of the baselines based on the public ACTIVE-3D-DET toolbox. We followed all KECOR training settings, using Adam as the optimizer, and a onecycle learning scheduler with an initial learning rate of 0.01. The batch size was set to 6, and each active round was trained for 40 epochs before proceeding to a new sampling round. We used one NVIDIA RTX A6000 to complete all experiments. The runtime for an experiment on KITTI and Waymo is approximately 5 and 81 GPU hours, respectively. The model embeddings $f^e$ used in our method are extracted from the second convolutional layer in the shared block of PV-RCNN.

## B  More Experimental Details

**DDFH in the KITTI Dataset.** In Fig. 6a, we present the performance of various AL methods in each active round. The number of point clouds in each active round is fixed, allowing us to compare the performance of models under conditions where they have seen the same number of scenes. Notably, KECOR's performance is below expectations given the same number of frames, indicating that KECOR does not effectively consider the diversity information of the scenes. In contrast, DDFH considers frame-level information to avoid redundant instances in similar scenes. The results show that DDFH has a significant advantage in each active round. We present more

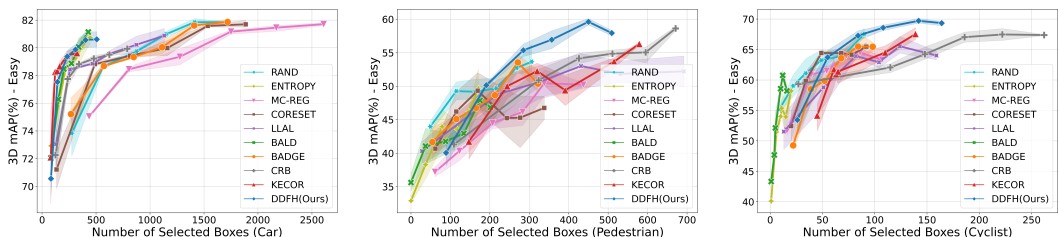

Figure 7: 3D mAP(%) of DDFH and the AL Baseline across various categories on the KITTI dataset at the moderate difficulty with PV-RCNN.

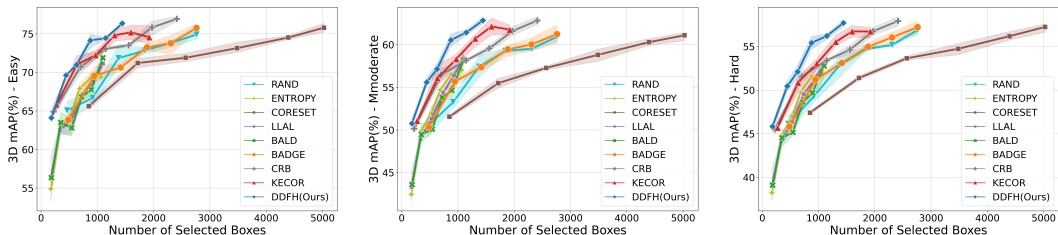

Figure 8: 3D mAP(%) of DDFH and AL baselines on the KITTI val split with SECOND.

comprehensive experimental results of DDFH on the KITTI Dataset in Fig. 7 and Fig. 8. The results in Fig. 7 indicate that DDFH with PV-RCNN has a significant advantage in all categories in KITTI, consistent with the results of Figure 4 in main text on SECOND. It is noteworthy that in the car category, some uncertainty-based methods achieve similar performance to DDFH with the same annotation cost. However, these methods fail to improve effectively in other categories, demonstrating DDFH's effectiveness in resource allocation and diversity. Fig. 8 also provides the trend of average 3D mAP for the one-stage model SECOND in different difficulties, consistent with PV-RCNN, outperforming SOTA methods in all difficulties. Further, in Table 1, we provide the performance of PV-RCNN trained on 100% labeled data, showing that DDFH's performance with only 1% of bounding box annotation is close to fully trained performance, even outperforming fully trained models in the pedestrian category.

**Ablation Study of Density Estimation.** We also test the stability and generalizability of DDFH through different density estimation methods and parameters. In Fig. 6b, we set different numbers of GMM components, specifically 1, 10 (DDFH Ours), 50, and 100. The results indicate that all experiments, except for 1 component, maintain similar effectiveness. In Fig. 6c, we use Kernel Density Estimation (KDE) to estimate the probability density and adjust different bandwidths to test the stability and generalizability of the DDFH. Silverman [3] and Scott [4] calculate bandwidth based on sample size. The results show that the performance of DDFH remains consistent and stable under different density estimation models and parameters. This is due to the distribution discrepancy focusing on distribution differences and novelty, rather than relying on highly accurate distribution estimates, thus providing sufficient robustness to noisy instances and estimation deviations.

# C Limitation

Considering that the distribution of objects in real environments is often uneven, common objects tend to occupy the majority (e.g. cars). This leads to the underestimation of less frequent categories when estimating informativeness. Therefore, the components DD, FH, and CB in DDFH reduce the impact of uneven distribution at different levels, decrease redundant annotations, and effectively balance minority categories. Although most real-world scenarios exhibit an uneven long-tail distribution, if specific situations lead to a dataset where object distribution is close to a uniform distribution, the effectiveness of DDFH might be limited due to the less apparent distribution differences. A possible solution is to incorporate indicators of uncertainty into DDFH, such as model instability, entropy, or the kernel coding rate combined with KECOR. This approach could address the mentioned limitation and is left for future research.

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
