# OpenReview forum: "Distribution Discrepancy and Feature Heterogeneity for Active 3D Object Detection"
_robot-learning.org/CoRL/2024/Conference — CoRL 2024_

### Official Review · Reviewer_jH4f · 2024-07-18
**Review of DDFH for Active 3D Object Detection**

**Originality:** 3
**Technical Quality:** 4
**Clarity Of Presentation:** 4
**Potential Impact:** 3
**Recommendation:** 4
**Confidence:** 3

**Review:**

The authors present a framework for active learning over segmented lidar data from the Waymo and KITTI datasets. The proposed method minimizes the cost of human annotations while maximizing prediction accuracy using frame-level heterogeneity, instance-level distribution discrepancy, and confidence balance for imbalances classification data.

The paper validated the proposed methodology against 12 state-of-the-art approaches and demonstrates improvements over all methods for birds-eye view and 3D mAP evaluation metric as well as minimizes the annotation cost compared to these methods. Overall, the authors have present their work clearly, concisely, and convincingly, and provide an ablation study to validate their choices. I would note that the authors do not discuss limitations of their work, or even future research directions. I would be interested to see these elements included. Other than some minor grammatical errors and adding this limitations section, I see no additional required edits.

**Quality Of The Limitations Section:**

1

**Questions For Rebuttal:**

Fix minor grammatical errors and add a limitations section.

**Robotics Focus:**

2

**Summary Of Paper:**

The paper proposes a metric for active learning to choose a lidar frame for human annotation to maximize informativeness of the new labeled data with respect to existing labeled data. The proposed method uses three metrics to estimate informativeness: instance-level distribution discrepency, frame-level feature homogeneity, and confidence balance.

**Summary Of Recommendation:**

Great presentation and compelling results. Overall an excellent paper.

---

### Official Review · Reviewer_Ybjb · 2024-07-25
**Review of Distribution Discrepancy and Feature Heterogeneity for Active 3D Object Detection**

**Originality:** 3
**Technical Quality:** 4
**Clarity Of Presentation:** 3
**Potential Impact:** 3
**Recommendation:** 3
**Confidence:** 4

**Review:**

Strength:
1.  The paper is well-written, easy to comprehend, and logically organized into sections.
2.  The figures effectively complement the text, facilitating easy understanding of the tasks.
3. The proposed approach is well desgined according to address the challenges of active learning for LiDAR-based 3D Object Detection
4.  The proposed approach outperforms existing state-of-the-art methods on both KITTI and Waymo dataset.

Weakness:
1. Clarification on Graph Interpretation: It is unclear why a specific number of maximum selected boxes was chosen for the graphs individually for each approach. Why not use a consistent number, such as 4000 selected boxes, across all approaches? From the plot, the reported approaches (for 4000) seem to saturate by the time they reach 4000 selected boxes. Using a consistent number of selected boxes, for all approaches would provide clearer comparisons on their efficacy. For example, DDFH, KECOR, and CRB show growth, but it is uncertain if they would saturate if 4000 boxes were used. Given that obtaining 4000 annotations is relatively insignificant compared to the total of 80000 annotations, using this number would enhance the comparability and clarity of the results.

2. In my opinion, the main table should also include the results of the PV-RCNN and SECOND approaches using the full dataset. This addition would provide a clearer understanding of how close AL methods are to the upper limit of performance.

3. The figure placements are a bit odd. In the main results section, Figure 5 is referred to on page 7 after Figure 3, and Figure 4 is referred to on page 8. Given that Figure 5 contains the main results, the figures should be rearranged for a smoother flow. Additionally, Figure 5 reports different results for two different datasets. The caption should clearly mention that parts (c) and (d) are for the KITTI dataset.

4. Could you provide some intuition for the results in Table 3? Transitioning from LB to CB improves the average mAP score due to a significant increase in the Cyclist category, but the performance on pedestrians decreases. Adding FH further decreases performance for both Pedestrians and Cyclists. Why do you think this happens? Similarly, adding DD results in a drop for Cyclists. Do you have any speculation on why this occurs? Additionally, I recommend presenting results for the combination of DD+FH+LB.

**Quality Of The Limitations Section:**

1

**Questions For Rebuttal:**

Please address the points listed under weaknesses, especially points 1 and 4.

**Robotics Focus:**

2

**Summary Of Paper:**

The paper introduces a novel active learning method called Distribution Discrepancy and Feature Heterogeneity (DDFH), which considers both geometric features and model embeddings to assess information from instance-level and frame-level perspectives. DDFH generates several scores, including discrepancy score, novelty score, correlation, and variance score. These multiple indicators are then aggregated using Quantile transform to provide a unified measure of informativeness. Extensive experiments demonstrate the benefits of the proposed approach, as it outperforms the current state-of-the-art methods on the KITTI and Waymo datasets.

**Summary Of Recommendation:**

I recommend a weak accept, as there is a definite use case for the proposed work. However, if there were an option for a borderline recommendation, I would have chosen that until clarity is provided for weaknesses 1 and 4.

---

### Official Review · Reviewer_x4Ww · 2024-07-28
**Review of submission 329**

**Originality:** 3
**Technical Quality:** 2
**Clarity Of Presentation:** 2
**Potential Impact:** 3
**Recommendation:** 3
**Confidence:** 3

**Review:**

The paper is well-organized and the main idea is easy to follow. Active learning is important to data efficiency, LiDAR-based 3D object detection is crucial for autonomous driving, and the ground truth bounding boxes are challenging to obtain in the real world. So the motivation is sound. The experiment is extensive, involving different datasets and comparing multiple recent baselines. I have some concerns about the method part.

- For the GMM-based distribution discrepancy, with the labeled GMM fitting $G_c^L$ and unlabeled GMM fitting $G_c^U$, how can you correctly align each component of $G_c^L$ and $G_c^U$ to find the distribution discrepancy correspondingly? i.e. in Eq (1), maybe the distribution of $\textbf{f}$ in $G_c^L$ is totally different from $G_c^U$ as they are based on different and misaligned Gaussian components. Also, how to dynamically specify the number of components in GMM fitting in the active learning process?

- For the feature heterogeneity, the covariance and correlation in Eq. (4) are for different dimensions of the mixed features of labeled  $F^L$ and unlabeled $F^U$, but there may be some noise correlation within labeled  $F^L$ or unlabeled $F^U$. Why not directly calculate the correlation between labeled  $F^L$ and unlabeled $F^U$ for each dimension? Also, Eq. (5) seems to be too heuristic and lacks math formulation, and I wonder if Mahalanobis distance between features can be adopted to measure heterogeneity. More theoretical and empirical justifications are needed.

- The Quantile Transform $\psi$ is not formally defined, when it is mentioned in Eq (3) and (6). It is expected to mathematically define the nonlinear transform $\psi$. Besides, the introduction of score normalization is not clear and confusing, especially the right part in Figure 2. For the left part of Figure 2, is the bounding box information also incorporated into the input feature of DDFH? If so, for the unlabeled samples, how do we obtain the input feature since there are no bounding boxes?

- It is not clear why LB is limited in 3D object detection and the formulation of CB seems to be trivially modified from LB. In the experiment, Table 3 shows that CB performs better than LB without DD and FH for imbalanced classes like Cyclist, but after DD and FH are added, it seems that the performance of imbalanced classes like Cyclist decreases even with CB. Does it mean that CB does not work well together with DD and FH? It is expected to show how LB works together with DD and FH.

- Some minors: Line 82 "Estimating ...[36] [37]" is not a complete sentence. Line 152, notation $N$ is not defined and seems to be confusing. The notation of $\hat{F}$ in Eq (4) is not consistent with $\tilde{F}$ on line 154-155.

**Quality Of The Limitations Section:**

2

**Questions For Rebuttal:**

See Review part.

**Robotics Focus:**

3

**Summary Of Paper:**

This paper proposes an active learning method for LiDAR-based 3D object detection, by enhancing instance distributional discrepancy and frame-feature heterogeneity. The final information score is normalized to find the best data to annotate in future training for the largest data efficiency. Experiments have been conducted on different datasets to evaluate the proposed results.

**Summary Of Recommendation:**

Concerns about the soundness of the proposed method, some incomplete experiments

---

### Author Rebuttal · Authors · 2024-08-08

We would like to thank all reviewers for their constructive and valuable comments on our work. We are also grateful for the meta-review, which summarizes the strengths of our work recognized by the reviewers.

Regarding the concerns and questions raised by the reviewers, we conducted several experiments and provided detailed results to address them. Please find these in our response to each review and in the new results presented in the attached PDF file. If there are any remaining concerns after reading our response, please let us know.

---

In the attached PDF, the following results are presented:

- Table 1: Full results on the KITTI dataset (including object detector PV-RCNN trained with 100% data) (**Ybjb**)
- Table 2: Full results on the KITTI dataset (including object detector SECOND trained with 100% data) (**Ybjb**)
- Table 3: Full results of the ablation study (including the combination of DD + FH + LB) (**x4Ww**, **Ybjb**)
- Figure 1: New results using SECOND as the object detector on the Waymo dataset
- Section A: More details on Quantile Transform (**x4Ww**)

---

### Decision · Program_Chairs · 2024-09-04

**Decision:**

Accept

**Comment:**

Summary: This paper proposes DDFH, an active learning method for LiDAR-based 3D object detection that reduces annotation costs by 56.3% by simultaneously considering geometric features and model embeddings, ensuring feature diversity and efficient learning with limited data, outperforming current state-of-the-art methods on the KITTI and Waymo datasets.

Strengths:
* Proposes a novel active learning method (DDFH) for LiDAR-based 3D object detection, addressing challenges by enhancing instance distributional discrepancy and frame-feature heterogeneity.
* Outperforms state-of-the-art methods on KITTI and Waymo datasets.
* Includes extensive experiments and comparisons with multiple recent baselines, validating the proposed approach.

Weakness:
* Some key aspects, such as GMM-based distribution discrepancy and feature heterogeneity calculations, are not clearly explained and need more theoretical and empirical justification.
* Inconsistencies in the number of maximum selected boxes across different approaches in graphs make comparisons less clear.
* The method's performance on imbalanced classes like Cyclist and Pedestrian shows inconsistencies, with certain combinations decreasing performance.
* The paper does not discuss potential limitations or future research directions.

----

Post rebuttal: The reviewers are happy with the authors' replies and have reached a consensus on accepting the paper.